# An Analysis of Transcriptomic Burden Identifies Biological Progression Roadmaps for Hematological Malignancies and Solid Tumors

**DOI:** 10.3390/biomedicines10112720

**Published:** 2022-10-27

**Authors:** Dashnamoorthy Ravi, Afshin Beheshti, Kristine Burgess, Athena Kritharis, Ying Chen, Andrew M. Evens, Biju Parekkadan

**Affiliations:** 1Rutgers Cancer Institute of New Jersey, Rutgers University, New Brunswick, NJ 08901, USA; 2KBR, Space Biosciences Division, National Aeronautical and Space Administration, Ames Research Center, Moffett Field, CA 94035, USA; 3Tufts Cummings School of Veterinary Medicine, Grafton, MA 01536, USA; 4Hematology Oncology, ICON Medical Affairs, Blue Bell, PA 19422, USA; 5Department of Biomedical Engineering, Rutgers, The State University of New Jersey, Piscataway, NJ 08854, USA

**Keywords:** tumor progression, transcriptomics, biological trajectory, lymphoma, leukemia, solid tumors

## Abstract

Biological paths of tumor progression are difficult to predict without time-series data. Using median shift and abacus transformation in the analysis of RNA sequencing data sets, natural patient stratifications were found based on their transcriptomic burden (TcB). Using gene-behavior analysis, TcB groups were evaluated further to discover biological courses of tumor progression. We found that solid tumors and hematological malignancies (n = 4179) share conserved biological patterns, and biological network complexity decreases at increasing TcB levels. An analysis of gene expression datasets including pediatric leukemia patients revealed TcB patterns with biological directionality and survival implications. A prospective interventional study with PI3K targeted therapy in canine lymphomas proved that directional biological responses are dynamic. To conclude, TcB-enriched biological mechanisms detected the existence of biological trajectories within tumors. Using this prognostic informative novel informatics method, which can be applied to tumor transcriptomes and progressive diseases inspires the design of progression-specific therapeutic approaches.

## 1. Introduction

Precision medicine guided by genomics is emerging as a realizable, though limited, approach to improve survival of cancer patients. Currently, most precision medicine strategies focus on targeting select mutational abnormalities with notable successes such as EGFR, c-met [1,2]. The effectiveness of precision medicine can be limited due to a lack of actionable targets, high mutation rates and the dynamically evolving signaling circuitry associated with oncogenic disease progression [3]. These lessons were reported from several genomics guided precision medicine based clinical trials, which includes, SHIVA, Molecular Screening for Cancer Treatment Optimization (MOSCATO-01), Copenhagen Prospective Personalized Oncology (CoPPO), MAST, PERMED 01, and PREDICT [4,5,6,7,8,9]. Despite improvement in progression free survival observed in a minority of patients, representing only 10% of the total patient population eligible to receive targeted therapies based on drug availability and drug-target matching criteria, a large number of cancer patients remain unable to benefit from precision medicine guided therapy [10,11,12,13].

A challenge in advancing precision oncology is the lack of knowledge about the chronological order of biological progression paths taken by tumors [3]. An imperative aspect of translational science is the generation of knowledge for precision medicine applications through the pooling and analysis of databases on the omics level. This information is available from academic and industrial entities funded by public and private funds [14]. Transcriptomics can provide a real-time overview of genome-wide RNA expression activity within a cell, essentially highlighting the principal biological activities dominating at a particular interval of time. Transcriptomics (microarray or RNA sequencing based) have provided tremendous biological insight into physiological states, drug effects in vitro and in vivo [15,16]. In contrast, tumor transcriptomes have been described as being quite noisy and less suitable for constructing meaningful systems level summaries of biology [17]. Despite several advancements in technology and sophisticated computational methods, charting the meaningful biological trajectory of tumor has not been well defined at the system-level [18].

Transcriptomic datasets obtained from heterogeneous mixtures of cells consisting of stochastic patterns of gene expression and variable mRNA abundance present a huge challenge for interpreting cancer biology. Physiological and pathological conditions, stress, drug perturbations, cell type differences and interactivity all affect gene expression patterns [19]. Transcriptomic analysis by gene regulatory networks, mathematical algorithms, neighborhood analysis, and artificial intelligence have all been used to determine a biological roadmap of tumor growth, but mostly resulted in identifying few clusters of biological classifiers among cancer patients [20,21]. The lack of longitudinal gene expression surveillance from the same patient is described as a significant limitation in reconstructing the biological path of tumor progression [20,21]. However, these analytical strategies suffered from setbacks due to their gene-centric nature and the lack of unbiased nonparametric techniques for defining a biological order among patient population. 

A similar problem of determining cell cycle progression trajectory from asynchronous cell population of different sizes and DNA contents existed before [22]. By applying the ergodic principle, flow cytometry finally resolved single cell snapshots obtained from asynchronous cell population into linearly arranged phases of the cell cycle [23]. An ergodic assumption states that the fraction of cells in a phase is equal to the proportion of time a single cell spends in that phase relative to the total cell cycle duration [23]. By applying a similar concept we rationalized that the transcriptome of an individual patient at a given interval represents proportion of time spent by tumor in that phase in relation to entire stages of tumor progression. Therefore, we developed a method providing an independent positional assignment for each patient through weighing their individual transcriptome and integrated gene behavior function across all disease phases. Essentially, these metrics constitute an abacus-like frame for the gene expression datasets in which patients are ordered by their overall increases in transcriptomic activity and their gene functions into progressive trajectory of biological complexity. The following partitioning of data and bioinformatic analysis defined the biological nature of tumor progression on a global level. Our analysis of 21 different tumor types composed of 4179 cancer patients provides a blueprint for predicting the causal path of disease progression and potential therapeutic targets. Finally, we provide evidence that this directional biological trajectory is present in malignancy through a prospective drug interventional study involving lymphoma in companion dogs. As part of this study, we leverage our prior comparative oncology investigational experience with canine lymphoma to facilitate human translational studies [24]. Such comparative oncology investigational approach is increasingly regarded as a valuable strategy for bridging the gap between basic and clinical research [25].

## 2. Materials and Methods

### 2.1. Tumor Transcriptomic Datasets

Meta-analyses of malignant tumors were performed with median normalized log_2_ transformed transcriptomic data sets retrieved from publications or public repositories as defined further. Cornell DLBCL dataset available for n = 75, downloaded from [26], inter-sample comparability of TPM counts was validated by comparing the stable expression of 12 housekeeping genes. Remaining dataset were retrieved as harmonized median normalized log_2_ transformed, this included, NCI DLBCL available from [27] was used in this analysis, original data matrix consisted n = 562 from multiple sources, of which n = 481 from NCI was used for this analysis. Other datasets were downloaded from C-Bioportal repository; as harmonized median normalized log_2_ transformed includes, Target-2018 Phase II, expression levels for 26,136 genes in 203 all cases (RNA-Seq RPKM), [28], AML aml_ohsu_2018, mRNA expression n = 451, mRNA expression (RNA Seq RPKM) [29]; Breast cancer-2012 (n = 1904) METABRIC, University of Cambridge mRNA expression (Illumina Human HT-3 v3 microarray) [30] and TCGA-2017 bladder cancer (n = 408) mRNA gene expression (RNA Seq V2 RSEM) [31]. Pediatric extracranial solid tumors (n = 657) and normal (n = 147) (RNA Seq, Illumina, TMM normalized, FPKM transformed available from Oncogenomics Expression Database (https://omics-oncogenomics.ccr.cancer.gov/cgi-bin/JK, accessed date on 12 November 2021) [32].

### 2.2. TcB Analysis Pipeline

Gene expression values were log_2_ transformed, sum of gene expression values by sample were first calculated, followed by calculation of median shift by each sample was performed and these values were designated as TcB. Then, gene median shift by individual gene was calculated. Median shift = (value-range minimum)/range. Data frame was then rearranged in increasing order of TcB or gene median shift, grouped by TcB as lowTcB (0–0.25), midTcB (0.375–0.625) and highTcB (0.75–1). Differential gene expression between groups by pairwise (edgeR or Dseq2) or within group by pairwise comparing against group median was performed using R and initial analysis was performed using Network analyst version 3.0 [33]. Significant genes (*p* < 0.0005), FDR 0.05 were used for pathway enrichment analysis by reactome, and gprofiler. GSEA enrichment option for analysis by multiple databases are included in R code. 

### 2.3. Availability of Computer Code and Algorithm

TcB calculations, transcriptomic ordering, grouping, significant genes and pathway enrichment analysis were written as R code and provided as Appendix B. 

### 2.4. Network and Heatmap Analysis

Network analysis of reactome enriched pathways were performed using EnrichmentMap tool [34] available from Cytoscape version 3.8.2 [35]. Heatmap analysis represented as row variances, hierarchical clustering based on one minus Pearson’s correlation by predefined groups, heatmap collapsing was performed based on group medians were performed using Morpheus available from https://software.broadinstitute.org/morpheus/ accessed on 12 November 2021.

### 2.5. Statistical Analysis

Significant genes by fold change > 1.25 (log_2_), or 0.25 fold for median shift transformed datasets, FDR and *p* < 0.005. Reactome enrichment FDR 0.05 and q-value cutoffs *p* < 0.0001. Quantitative rendering represented as box plots, ribbon plots, lollipop plots, density plots and parallel index plots were generated using OriginPro 2021 (OriginLab Corporation, Northampton, MA, USA). 

### 2.6. Canine Lymphoma Study

Canine B cell lymphoma subjects (n = 10) were enrolled and treated with BKM120 in an IRB and IACUC approved veterinary clinical study, performed at Tufts Cummings School of Veterinary Medicine, Grafton, MA. All experiments pertaining to canine clinical study were performed in accordance with relevant guidelines and regulations. Canine subjects received BKM120 2.25 mg/kg orally for 28 consecutive days. BKM120 provided as gift from Novartis. Analysis for tumor response were performed by direct tumor measurement or through the use of CT imaging. Furthermore, fine needle aspirations of the lymphoma nodes were collected on Days 0, 7 and 21 and utilized for RNA isolation and gene expression analysis. Routine blood and urine analysis were performed for toxicity evaluation and as part of standard clinical assessment. RNA isolation was performed using RNeasy mini kit (Qiagen, Germantown, MD, USA) and transcriptomic analysis with canine array. We performed unbiased assessment to determine pertinent biological pathways associated with treatment response as has been shown before [16,36].

### 2.7. DNA, RNA and Protein Synthesis Assay

DNA synthesis by EZClick EdU cell proliferation kit (#K946), RNA synthesis by EZClick Global RNA synthesis assay kit (#K718), Protein synthesis by EZClick Global Protein synthesis assay kit (#K459) were purchased from Biovision (Milpitas, CA, USA), assays were performed following the instructions supplied by the manufacturer, and analyzed by flow cytometry as described before [36].

### 2.8. Cell Cycle Analysis

Fluorescent cell cycle reporter, cell cycle green/red lentiviral EF1a, Puromycin (Sartorius, Bohemia, NY, USA) was transduced in SUDHL-4 cells, as described before [37]. Following puromycin selection, transfected SUDHL-4 cells were plated at 50,000 cells per well in 96 black clear bottom well plate, pre-coated with retronectin and continuously imaged every 2 hours following the release from BKM120, using Incucyte Zoom (Sartorius, Bohemia, NY, USA).

## 3. Results

### 3.1. Ordering Gene Expression Signatures by TcB

Approximately 22,000 protein coding genes are expressed at varying levels in the transcriptome of individual cell or patient at any given time. Considering that malignancy is a progressive disease, collection of transcriptomic signatures of various patients represents a biologically disordered dataset. Moreover, malignancies are well known to progress in a manner marked by unrestrained proliferation, which is accompanied by increases in DNA activity, which can be directly linked to increased global transcription [38]. Following these established biological principles, we developed a strategy for rearranging biologically disordered transcriptomes into increasing levels of transcriptional activity, henceforth termed transcriptomic burden (TcB). Evaluating patients with different TcB proved fruitful in identifying biological trajectories associated with tumor progression.

In the first step, we initially re-organize patients based on the median gene expression levels across all patients. In essence, this stratifies patients based on their overall RNA activity (which we refer to a subject’s TcB) relative to the total population under study. In practice, we first calculate the sum of normalized RNA expression values for every transcribed gene by each patient, (represented in each column) (Figure 1A). Then, standardized TcB values were calculated (using sum of gene expression value series, as numerical shift, a directional measure that assigns positional identities to each patient as numbers between 0 and 1 (Figure 1A). 

Once data was re-ordered around patient TcB, a second informatic process was performed to then evaluate individual genes within the different TcB groups. Again, we used median shift for each gene across all patients to clarify the direction of individual gene function in the disease, as “gene shift” (Figure 1A). This step reordered the transcriptomic data-frame by increasing order of TcB and gene shift, using these metrics as abacus like frames, resulting in a linearly aligned dataset as summarized in Figure 1A. Because 0.5 represents the midpoint on this scale, patients whose TcB shift is between 0 and 1 collectively represent the order of patients based on the progression of their transcriptome from low to high TcB, as shown in Figure 1A,B. Further through systematic comparison of transcriptomic data segmented based on low, mid, and high TcB it is possible to determine progressive biological changes occurring in tumors based on differential gene expression analysis, as shown in Figure 1B.

This TcB analysis method was applied to a previously published diffuse large B cell lymphoma (DLBCL) transcriptomic data as training set consisting of n = 75 patients [26]. The original data set consisted normalized RNA counts reported as TPM (transcript per million) was analyzed as an unordered transcriptome of patients by heat map (Figure 2A). This dataset, transformed as a scatter plot by gene function, illustrated a randomly distributed gene activity across the entire spectrum of the disease (Figure 2A). TcB analysis was then performed on the transcriptomic datasets of 75 DLBCL patients represented by 19,734 genes. Patients were stratified by median shift cutoffs (0–0.25, 0.375–0.625 and 0.75–1, (n = 49), respectively, as low, mid and high TcB) with patients remaining outside of these cutoffs were excluded. TcB analysis showed that the original dataset became resolvable into different populations as shown in scatter and density plot (Figure 2B). With increasing TcB, we noticed that gene activity spread (measured as gene shift) gradually began to condense, shown in the scatter plot (Figure 2B). 

These TcB groups were then used for statistical comparison for differential gene expression across and within groups. There were 1,136 genes that showed significant up or downregulation between TcB groups (*p* < 0.0005, FDR 0.05), shown as heatmap (Figure 2C) (see Appendix A). By using Gprofiler and reactome pathway enrichment analysis tools, we identified and grouped significant genes by their corresponding biological functions (see Appendix A). We then organized these pathways into interactive biological networks using cytoscape. In a surprising finding, DLBCL patients with low TcB had the most complex network of interconnected pathways as opposed to their mid or high TcB counterparts (Figure 2D–F). Biological enrichment in low TcB included signaling, translation, and metabolic functions (Figure 2D and Appendix A). In contrast, a predominance of transcriptional regulation (gene expression) and chromatin modification, with no apparent enrichment for signaling, translational, or metabolic pathways were observed in high TcB groups. Mid TcB groups also showed transcriptional regulation and chromatin modification too, along with cell cycle pathways, transcription regulation, and receptor tyrosine kinase signaling (Figure 2E,F and Appendix A). Considering that DLBCL is classified by B lymphocyte subtypes (germinal center subtype or activated B cell subtype) or molecular subtypes (myc-bcl-2 double expresser), we assessed if the enriched biological processes were specific to disease subtypes. However, our results indicate that biological enrichments identified through TcB ordering showed considerable homogeneity across DLBCLs, irrespective of cell of origin or molecular subtypes (Appendix A). Together, these results, based on the comparison of transcriptomic changes across the full spectrum of DLBCL, suggest that tumor progression can be predictably described as gradual changes in biological functions. 

### 3.2. TcB Stratification Identifies Conserved Biological Patterns across Tumors

TcB stratification was applied to multiple tumor transcriptomic datasets to further verify the approach. Previously published log_2_ transformed datasets representing DLBCL (from National Cancer Institute-NCI, n = 481), [27], acute myeloid leukemia (AML from Oregon Health & Sciences University, n = 451) [29], acute lymphoblastic leukemia (ALL from NCI Target-2018, n = 203) [28], breast cancer (METABRIC, n = 1904) [30] and bladder cancer (TCGA, n = 408) [31] were analyzed. The biological pathways identified for all tumors were then collapsed to reveal higher order processes that drive cellular function, as shown in Appendix A, and the pattern of changes categorized by TcB groups are shown in heatmaps represented in Figure 3A. When tracking the overall behavior of these higher-order functions by averaging expression of genes representing each pathway, we found that every single pathway followed a uniform pattern of changes in all tumors, except in AMLs that originates from hematopoietic stem cells (Figure 3B). These results clearly revealed that protein translation, electron transport chain and citric acid cycle, transcription, the cell cycle and ECM are enriched higher-order biological functions that were consistently resolved by TcB. Due to our cutoff rule, the analysis was performed on fractions of the original populations (Figure 3C) though still considered notable sample sizes with significance was stringently determined at the genomic level. These results suggested conservation of these processes across various tumor types could be illuminated by TcB enrichment and analysis.

### 3.3. Correlating Gene Functions and TcBs in Pediatric Solid Tumor Progression

Datasets analyzed thus far represented unique cancer types and did not include normal tissue controls. TcB changes may be a physiological function irrespective of pathological status; therefore, we evaluated higher order processes revealed by TcB analysis with normal controls to see if the data contained artefacts. For this study, we analyzed a recently published transcriptomic dataset consisting of 657 pediatric extracranial tumors with 14 cancer diagnosis types matched to 147 normal tissues [32]. Using TcB analysis, the entire panel was first examined, and datasets representing each tumor type were evaluated separately. Results from the full panel analysis showed that normal tissues had lower TcB values when compared to 14 different types of tumors (Figure 4A). These results suggested that global transcriptomic activity generally increases in tumors. Unlike other tumors, the higher order biological enrichment detected in this panel was restricted to cytokine activity and ECM genes that increased as TcB increased (Figure 4B and Appendix A). Transcription regulation genes decreased in normal tissue as TcB increased (Figure 4B and Appendix A). Further examining the log_2_ mean of combined gene expression of the detected higher order processes compared to normal tissue as a baseline, it became evident that cytokine genes are highly expressed in desmoplastic round cell tumors, melanoma, hepatoblastoma, osteosarcoma, alvelolar soft part sarcoma, and teratoma (Figure 4C). Most tumors, except yolk sac tumors, neuroblastomas, and Ewings sarcomas, expressed higher levels of ECM genes than normal tissue (Figure 4C). Similarly, with the exception of yolk sac and Wilms tumors, the expression of genes associated with transcription was reduced in most tumors (Figure 4C). 

Datasets containing n > 75 from this transcriptomic panel consisting of 14 different tumors and normal were then selected and analyzed individually. The transcriptomic dataset used for individual TcB calculations and biological analysis included Ewings sarcoma (n = 98), neuroblastoma (n = 227), normal (n = 104), osteosarcoma (n = 94) and rhabdomyosarcoma (n = 122). As shown by the heatmap of higher order enrichment analysis, gene expression of cell cycle, transcription and translation was decreasing with TcB increases only in tumors, but not in normal tissues (Figure 4D and Appendix A). The results from individual TcB analysis of pediatric tumors and normal tissues indicates that the patterns of changes represented by cell cycle, transcription and translation identified from previous analyses of solid tumors and hematological malignancies (Figure 3) as higher order enrichments and are unlikely to be an artifact.

We consistently observed declining patterns of cell cycle gene expression along with TcB increases in our analysis, but this is a counterintuitive biological phenomenon for tumor progression. Therefore, we sort to resolve this conundrum through identifying the biological nature of genes that became reduced with higher TcB. We therefore, compared correlation between TcBs and expression values of each gene across the entire dataset as a next step in our assessment. We identified the top 50 significant genes (*p* < 0.0001), by positive (r > 0.75) and negative (r < −0.75), by Spearman’s rank correlation from individual and entire panel analysis of Ewings sarcoma, neuroblastoma, normal, osteosarcoma and rhabdomyosarcoma. TcB clustering analysis of these top 50 genes identified two clusters, with immune function (represented by C1QA, C1QB, C1QC, CD14, CD68, CD74, FCER1G, IFI30, LGALS9, TYROBP, and HLA-DMA) as upregulated (Appendix A). Furthermore, these tumors also showed significant decreases (log_2_ FC, −10 to −5, highTcB versus lowTcB) in the expression of genes that encode for histone proteins (Figure 4D and Appendix A). This unusual feature of histone complex gene expression declining from lowTcB (log_2_, 4.5–7.56) with higher TcB falling below normal tissues were observed in these tumors (Figure 4D). In non-transformed human cells, histone gene expression and biosynthesis of 400 million histone proteins are tightly coupled with DNA replication and cell cycle, and essential for cell survival [39]. However, histone levels and expression are reported as inversely correlated with overall transcriptional rate in Drosophila [40]. Taken together, this observation of diminished histone gene expression via TcB correlation provides novel dimension into pathological basis of disease biology in these tumors, and warrants further proteome-level validations. Overall, by comparing the transcriptomes of tumor and normal we conclude that TcB-based biological predictions are not arbitrary, but as aligned with the nature of malignancy.

### 3.4. Charting the Biological Roadmap of Malignant Progression in Pediatric ALL 

Our next goal was to determine whether TcB shifts and biological patterns uncovered by this method were associated with progressive steps in malignancy. To this end, we required a study where 2 samples were taken from the same patient over disease progression. The Target 2018 ALL study fit this paired patient and time-varying sample collection criteria. This ALL study analyzed transcriptomic data from patients with initial and relapse diagnoses [28]. TcB analysis was performed among these patients with clinically defined relapse. Of the 41 patients analyzed, 24 developed a shift in TcB (from low to mid or higher) and 17 showed no such shift (remained at the same level) between diagnoses. In our analysis of this group for higher order biological pathway behaviors, we found that log_2_ expression profiles of genes involved in mitochondrial translation, TCA/ETC, cell cycle and transcription declined as TcB increased (Figure 5A). In contrast, ECM genes increased along with TcB (Figure 5A). Analysis of correlations by TcB shifts from repeated ALL samples collected from same patient at different intervals showed that patterns of TcB shifting from low to mid, and mid to high TcB shift values naturally occur with disease progression (Figure 5B). Results from these findings suggested that an increase in TcB alone could be an independent characteristic feature of ALL disease progression (Figure 5B). 

To further explore the biological directionality of this disease, we examined gene expression changes related to higher order processes in ALL, comparing initial versus relapse samples. The groups were again parsed into subsets that had a TcB shift between sample time and ones that did not. In quantitative directional analysis by lollipop plot, log_2_ FC genes (initial vs. relapse) associated with ECM progressive increase and appear between mid-to-high TcB or low-to-high TcB shifts (Figure 5C). Conversely, log_2_ FC of genes representing transcription, cell cycle, TCA/ETC, and mitochondrial translation (relapse vs. initial) was negatively regulated with TcB shifts, but not in ALL without TcB shifts (Figure 5C). Since these biological changes and patterns were consistent observed in most tumors and seem to exhibit directional activity in ALL, we conclude that genes constituted within higher order biological functions are likely to dictate disease progression in ALL. Together, these findings suggest TcBs and higher-order biological functions are indicators of directional properties in tumor progression (summarized (Figure 5D) and can have prognostic insight. 

### 3.5. Dynamics of TcB Ordered Biological Functions

Understanding the dynamics of TcB enriched higher order processes in the context of tumor progression is crucial notably because it is know that increases in ribosomal biogenesis and translation in G1, transcription in G2, reduction in transcription at M phase are related with cellular progression through cell cycle [41]. Cancer is characterized by disordered proliferation, and the biological features found within different subsets of TcB tumors as well as the features found in all tumor categories are likely to suggest a stepwise dysregulation of the cell cycle involving higher-order processes as the tumor progresses. Therefore, we attempted to determine the dynamics of these properties in cell culture models. Our analysis of the Broad Institute CCLE panel, which consists of 1527 cells lines, and other tumor panels did not resolve using the TcB method. TcBs were found to be skewed towards left, suggesting that most cell lines show greater levels of transcriptional activity than natural tumors, and perhaps reached a stable state during cell culture (Appendix A). As a result, evaluation of biological trajectory of tumor progression using cell culture models may prove challenging for validating our observations derived from the analysis of tumor transcriptomes. 

Given limitations of cell line analysis, we investigated the directional features of these higher order biological functions by adopting a natural disease, canine lymphoma. A longitudinal in vivo study of the tumor transcriptome at three time points was conducted using a phosphoinositide-3 kinase (PI3K) inhibitor, BKM120, as targeted drug (Figure 6A). BKM120 blocks the oncogenic PI3K signaling mechanism that plays an important role in metabolism and ribosomal biogenesis [42]. Additionally, we and others have previously reported that PI3K signaling is dysregulated in canine lymphoma [24,43]. With PI3K targeting, we rationalized that blocking higher-order functionality associated with low TcB would allow TcB and the properties of higher order biological functions to be reset. A total of ten dogs with B cell lymphoma were enrolled in this trial, and six of them completed the evaluation of BKM120, a PI3K targeted inhibitory therapy that was administered orally in cycles of 28 consecutive days, as summarized in Figure 6A. This evaluation, while lacking power for population-level outcomes, followed Simon’s “minimax” design due to the smaller sample size and lower subject enrollment in this large animal study [44]. The data gathered was therefore deemed adequate to gather reliable biological insights. Only 6 of the 10 dogs with lymphoma in this study completed at least one treatment cycle. Clinical characteristics, including breed, age, diagnosis, stage, treatment history, toxicity, and outcome, are summarized in Figure 6B. Based on peripherally measurable lesions as determined by consensus criteria defined by veterinary cooperative oncology group [45], 2/6 dogs treated with BKM120 showed partial response with more than 30% decrease in tumor volume; 1 dog with progressive disease, and 3 dogs remained with stable disease. Two of the six dogs that showed partial response had persistent blood glucose levels and were withdrawn before completing five cycles of BKM120 treatment in 28 days; the remaining four were removed because of lack of improvement or because the physician determined they had reached the end of their lives.

The transcriptomic assessments were performed on tumor biopsy samples collected at the time of diagnosis, 1 week and 3 weeks after treatment, and TcB estimation was performed using the dataset (n = 18). Results show that the TcB values of canine lymphoma subjects CL#1, CL#6 remained unchanged, while CL#2, CL#5 showed significant decreases over the course of 3 weeks with BKM120 (Figure 6C). By applying the median shift transformation, the entire canine transcriptome (represented by n = 30,311 genes) is scaled uniformly, so that global changes in gene function may be directly compared between different time intervals. The kernel density plot of these results clearly indicates that TcB declines gradually after 3 weeks of BKM120 treatment. The declining TcB effect was modest in CL#1, 2 & 6, which had stable or progressive disease (Figure 6B,C). Of 6 dogs, 2 had partial response showed rapid decline in gene expression within the first week after being treated with BKM120 (Figure 6B,C). Furthermore, CL#5, the subject with the most elevated gene expression levels, also had more dramatic reduction in gene activity by week 3, though with advanced stage and relapsed disease, this subject also died by week 3 (Figure 6B,C). As TcBs and gene shifts are compared, both net transcriptional activity (as measured by TcB) and gene function (as measured by gene shift) responded to therapy through downward shift (Figure 6D). Drug treatment affected expression of genes associated with TCA, ribosome/translation, and cell cycle which became gradually upregulated, while genes representing ECM became gradually downregulated over 3 weeks of treatment (for entire population) with BKM120, as shown by lollipop plot mimicking low TcB state in Figure 6E. By comparing the observed biological pattern reversal and TcB decreases in BKM120 treated canine lymphoma (Figure 6E), it becomes clear that biological functions associated with disease progression follow reversible chronological patterns. Based on the expression profile changes of genes representing higher-order biological processes, each canine subject shows progressive upregulation of genes involved in TCA, ribosome/translation, and cell cycle functions in all of them, with the exception of CL#5 with terminal disease pretreatment and post week 1. CL#5 showed an entirely dominance of genes involved in ECM that reflected a fully developed disease state (Figure 6F), as supported from the predicted progression pattern of human ALL patients (Figure 5E). By comparing these canine results to the generalized patterns observed in human DLBCL tumors, we conclude that these higher order biological functions both exhibit orderly responses and are reversible features. The findings were further confirmed by in vitro measurements of replication, transcription, and translation activities using SUDHL-4 (DLBCL cells) and BKM120 treatment (Figure 7A). The results indicate that translational activity, measured as global protein synthesis, is decreased with BKM120 treatment, which recovers by day 1 after drug release in SUDHL4 cells (Figure 7). Following the initial lag in protein synthesis in SUDHL4 cells, show a delayed global DNA and RNA synthesis (days 1–2) coinciding with delayed G2 recovery, following BKM120 treatment release (Figure 7A,B). We conclude based on our observations of BKM120 treatment in canine lymphoma and experiments using SUDHL4, as well as our predictions based on TcB analysis of human tumors, that the progression of malignant tumours can be characterized by a sequence of transcriptomic changes relating to protein transcription, replication, and translation.

## 4. Discussion

Cancer is defined as an uncontrolled growth of body cells that spreads to other parts of the body (source NCI/Cancer.gov) 12 November 2021. Although cancers arise from many different types of cells, the biological principles governing malignant progression seemingly may have common paths. Clinical observations indicate that malignancies, of the same type such as in lymphomas, can be indolent, turn aggressive, show progressive features or undergo cycles of dormant and active states [46]. Presently, there is no direct biological explanation for how these uncontrollable cells manifest an uneven proliferation rate [46]. Although mutation drifts and progressive genomic alterations at the molecular level are recognized in evolving tumors [47,48], the longitudinal biology behind tumor progression is still largely enigmatic. Transcriptome datasets are conventionally analyzed for significant genes or pathways using many different methods for computing differential gene expression have provided interesting seminal findings regarding genetic mutations and ex vivo drug sensitivity [26,27,28,29,30,31]. Yet, these studies did not attempt to generate global biological summaries because there existed no methodologies for linearizing the datasets for unbiased exploitation of all the available information. 

Several studies have attempted to unravel the biological progression roadmaps in cancer. Based on the evolution of genetic changes, Fearon and Vogelstein initially proposed a linear tumor progression sequence [49]. Many studies followed this approach by aligning gene expression datasets with mutational events in order to estimate temporal biological patterns in tumor progression [48,50,51]. Nevertheless, genomic datasets from cross-sectional tumor collection can include samples from unknown disease states, treatment status, environmental exposures, etc., which exhibit tremendous mutation heterogeneity and varying loads of mutational burdens. Therefore, aligning mutations with temporal biological order is considered a weak strategy [52]. Despite mutation assessments being considered sufficient guides for determining and administering effective cures, Vogelstein asserts that a crucial need in basic cancer research is a better understanding of the biological pathway trajectories [52]. There have been further models using probabilistic or Bayesian networks. In these models, genes in one pathway become parents of all genes in the next, and parental genes tied by mutation were integrated into the probability model [53,54]. Lastly, progression at the pathway level was inferred from a priori gene assignment, but only when the pathway had many gene sets. However, none of these approaches were successful in identifying homogenous biological trajectories among cancers, which led to the conclusion that cancer progression is non-linear. Major drawbacks of these approaches include the inability to analyze transcriptomic patterns unbiasedly and the lack of biologically appropriate hypotheses for identifying biological trajectories. Our strategy assumes that transcriptional complexity will continue to increase with tumor progression and the transcriptomic burden will increase. DNA levels in a cell remain relatively constant and can be synthesized in a short period of time. However, RNA levels are higher, so cells need longer periods of time to synthesize enough RNA to divide. Malignant cells may proliferate more rapidly when this constraint is removed, and their RNA content must gradually increase. Our results illustrate this generalized concept with several lines of evidence, including TcB shift from low to high-TCB in ALL, reversibility in canine lymphoma, and transcriptional lag in SUDHL4 cells with BKM120 treatment, all indicative of cancer progression. 

Transcriptomic analysis through a TcB strategy of linearizing tumor transcriptomes uncovered progressive biological features in tumors, laying the groundwork for more comprehensive analyses. Overall, the results from TcB based analysis support a homogeneity in tumors in terms of higher-order biological choreographers of malignant cell proliferation, whereas the signaling pathways involved in promoting such growth show heterogeneity. Interestingly, the translation (ribogenesis), replication, and transcription that were identified from our TcB analysis also correspond to central dogmatic principles which intersect progression fates of cell cycle. In this respect, our identification of these central dogmas being progressive aligned by TcB as in the same order of the cell cycle path defines biological mechanisms for tumor progression. Our findings also support the theory of embryonic reversal-based definition of malignant progression [55,56,57] in that changes observed in translation (ribogenesis), replication, transcription, and ECM in primary human tumors followed the opposite directions of higher-order processes occurring in human embryonic development [58]. TcB rendering of progressive network features of DLBCL, ALL, and pediatric solid tumors, as well as BKM120 treatment responses in canine lymphomas (see Appendix A), suggests that biological networks in tumors show continuous changes that are impressive and with predictable pattern. As TcB increases, biological networks that include signaling pathways, ribosomal biogenesis, translation, transcription, and cell cycle remain interconnected across low and mid TcB in DLBCL and ALL (Appendix A). In contrast, when the tumor transcriptome is enriched with high TcB, biological networks exhibit predominantly ECM reorganization, demonstrating that fully progressed tumors exhibit a biologically distinct state (Appendix A). Additionally, we found that, when the transcriptomic data was combined with the mutation profiles available for each patient in bladder cancer, a linear alignment by TcB also predicted that the mutations exhibited predictable evolutionary patterns (Appendix A). Mutation rates increased from low to mid-TcB from 300 to 420/patient but declined at high TcB (Appendix A). The number of genes mutated at low TcB (2000) dramatically increased to >8000 from mid to high TcB, which is another indication of tumor progression (Appendix A). A word cloud-based analysis of genes frequently altered in bladder cancers revealed that the TTN gene is most frequently altered, with RB1 and CST5 highly altered in low-TcBs but absent at higher-TcBs, p53 and RP11 alterations evident between middle and high-TcBs (Appendix A). In contrast, MUC16 and KMT2D steadily increased from low-TcB and became dominant at high-TcB (Appendix A). Furthermore, the number and frequency of gene modifications increase with a TcB trajectory, further validating the TcB methodology. 

Our TcB transformation and biological learning strategy has limitations, including a stringent cutoff that eliminated portions of patient for gene-centric analysis. The method must be further developed to accommodate samples representing continuous gradients in transcriptomics. Additionally, samples that do not represent the entire spectrum of tumor progression could skew the results. As an example, our analysis of transcriptomic datasets for head and neck cancers, lung adenocarcinomas, and colon carcinomas detected TcB shifts in either low or high levels, suggesting these samples could represent either early or advanced biological conditions, meaning biological trajectory extraction is not possible. Our analysis pipeline included curation of data that eliminated ambiguous biological processes, such as “disease” or “development”, since our intent was to illustrate and collapse enrichments based on biochemical activities directly related to cell proliferation. Several bioinformatic tools are required in order to render biological trajectory data, which is limited to the annotations and interactions defined within the databases. Future directions for improving this methodological strategy include procuring appropriate datasets through large prospective trials, integration of clinical outcome and multi-omic datasets, along with a diagnostic or drug-controlled decision making based on the information of TcB analyses to further solidify its use in actionable cancer care. 

## 5. Conclusions

Precision medicine requires high resolution biological insights to be able to predict tumor attitude and prescribe multifaceted targeted therapeutic strategies for effective cancer treatment [10]. Clinical failures frequently observed with precision oncology suggests progressive changes in biological network evolution as potential “Charlotte Web” factors that could facilitate evading and impeding the therapeutic efficacies of targeted drugs. To acquire further high-resolution biological perspectives into tumor progression, it would be necessary to integrate TcB analysis with transcriptomic data (such as bulk sequencing, single cell transcriptomics, etc.), multi-omics (such as metabolomics and proteomics) to generate a dynamic and multi-dimensional biological roadmap of tumor progression corresponding to each tumor type over time. Detailed rendering of TcB derived network dynamics as outlined in this study and defining combination therapies with integrated prediction of future biological states will have promising potential in precision oncology.

## Figures and Tables

**Figure 1 biomedicines-10-02720-f001:**
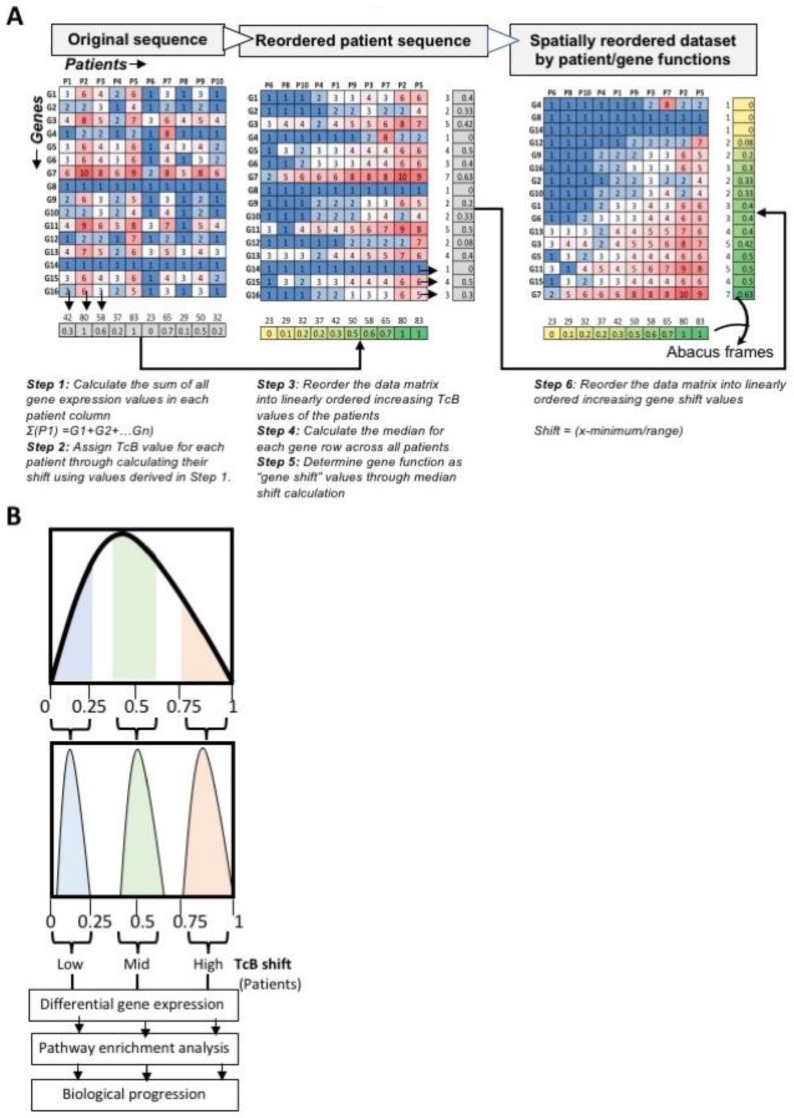
**Deconvolution of stochastically ordered bulk RNAseq reads from transcriptomic datasets by abacus ordering strategy.** (**A**) Schematic illustration of sequential steps involved in the calculation of transcriptomic burden (TcB) and the transformation of stochastically ordered gene expression data into linearly ordered dataset. (**B**) Theoretical distribution of gene expression and resolution by TcB with data partitioning criteria for patient grouping for downstream biological pathway enrichment analysis for discovering progressive patterns in gene expression.

**Figure 2 biomedicines-10-02720-f002:**
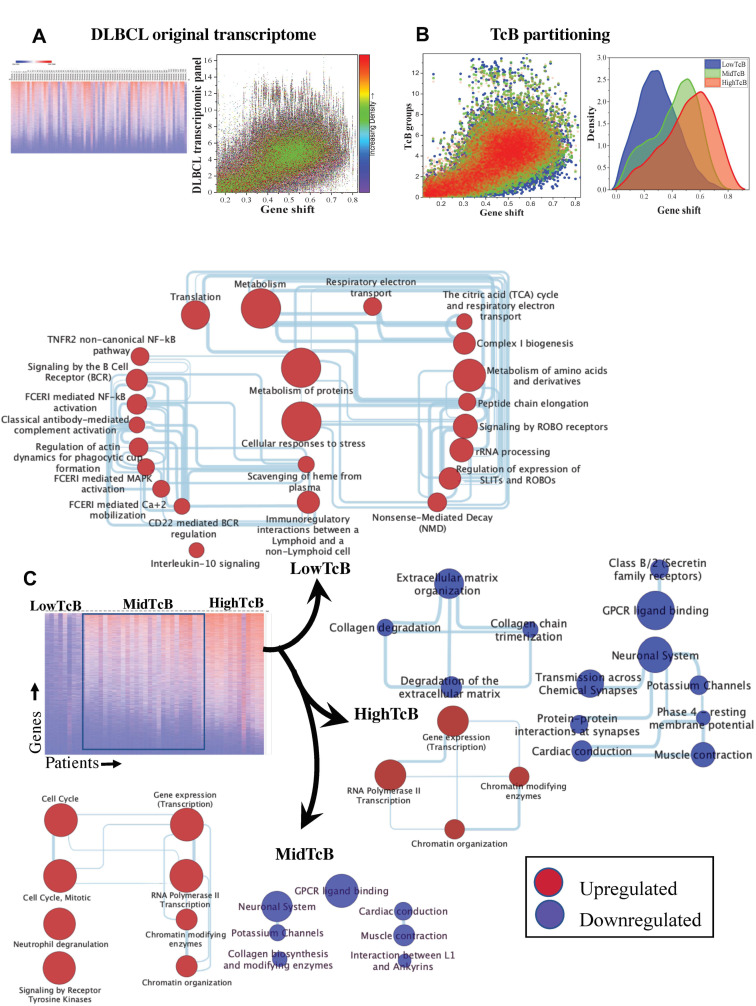
**DLBCL transcriptome analysis for biological progression.** (**A**) Heatmap of Cornell DLBCL gene expression original matrix (patients and transcripts) and scatter plot of TPM (transcript per million) counts (n = 75) vs. gene shift. (**B**) A scatter plot shows consolidation of ergodically distributed gene expression patterns vs. gene shift occurring as TcB increases, and resolution by density plot showing average TPM counts by ascending TcB series and delineated DLBCL patients into distinct subgroups. (**C**) Heatmap of DLBCL gene expression matrix sorted by TcB subgroups. Network representations of enriched biological pathways based on significant differences in gene expression between TcB groups indicate a reduction in biological complexity occurs with TcB increase in DLBCL patients.

**Figure 3 biomedicines-10-02720-f003:**
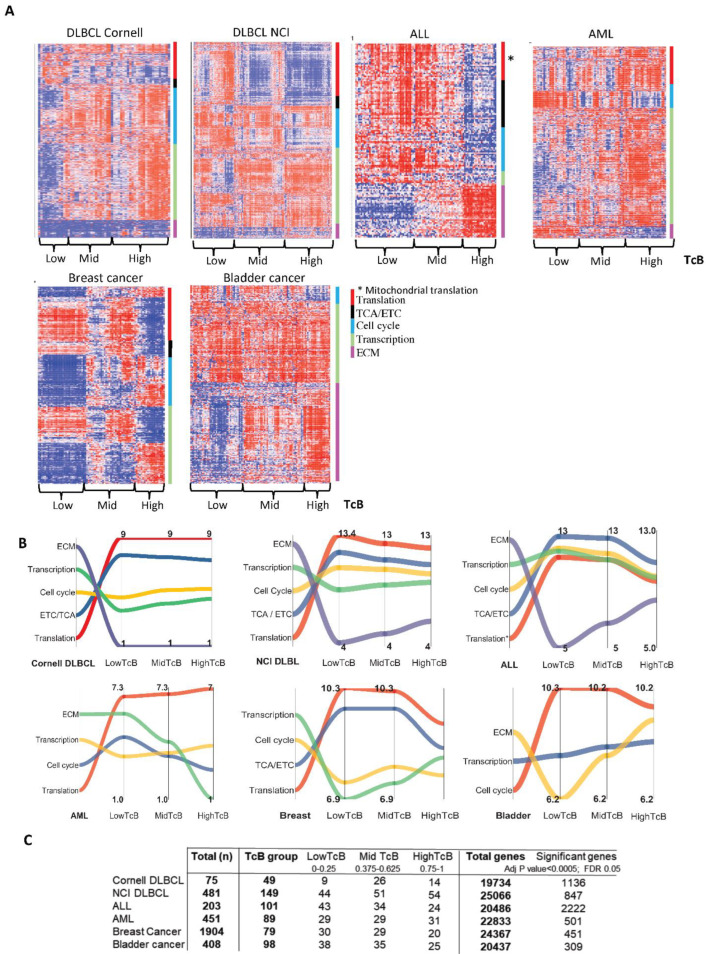
**Homogeneous biological patterns in malignancies identified by TcB analysis.** (**A**) The heat map represents higher order biological processes derived from collapsing reactome-enriched biological pathways determined from TcB subgroup analysis of independently analyzed tumor transcriptomes. A blue-to-red gradient indicates row variance. (**B**) Ribbon graphs of average log_2_ values of gene expression values for genes represented in higher-order processes shown in (**A**). (**C**) Summary of the numbers of patients/genes in the original tumor transcriptomic datasets and the TcB partitioned groups.

**Figure 4 biomedicines-10-02720-f004:**
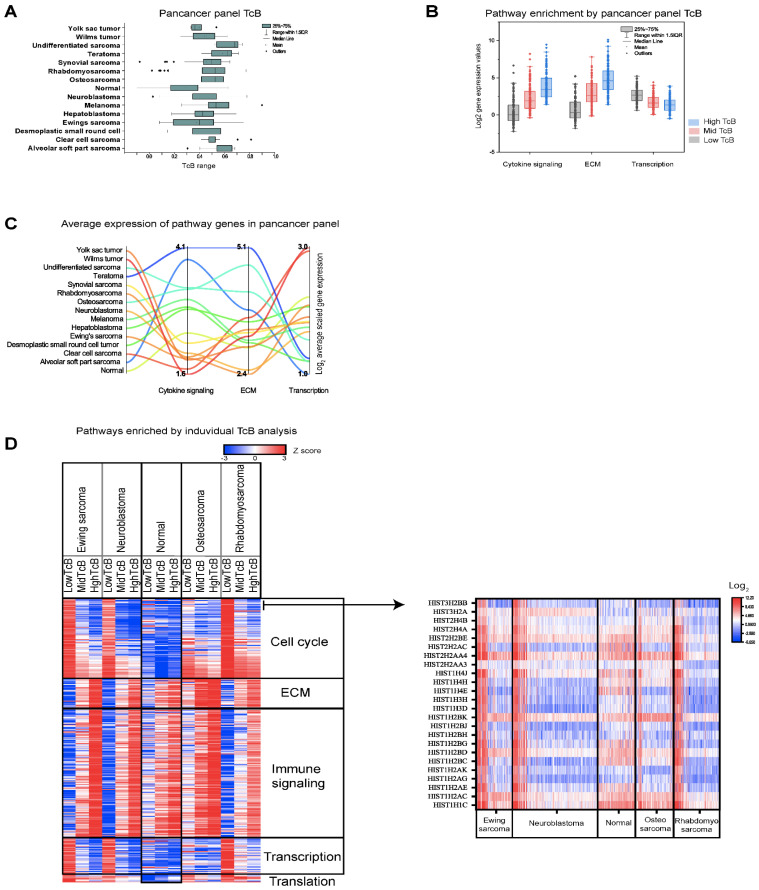
**Biological progression patterns in pediatric extracranial solid tumor and normal tissue panel.** (**A**) Bar graph represents values calculated from a transcriptomic dataset consisting of multiple pediatric solid tumors (*y*-axis) show that median TcB values (*x*-axis) of tumors are higher than normal tissues. (**B**) Bar graph illustrates median log_2_ expression levels (*y*-axis) of genes representing higher order processes (*x*-axis) determined from a pan pediatric solid tumor and normal tissue panel. (**C**) Summary of gene-expression differences between tumors and normal tissues (median log_2_), representing changes in higher-order processes. (**D**) Heatmap of higher-order biological processes enriched by independent TcB calculations based on the tumor or normal tissue type indicates unique biological properties are observable in tumors. An arrow points to the heatmap (right), which reveals that declines in histone complex gene expression as correlating feature in changes observed with cell cycle gene expression with TcB increases.

**Figure 5 biomedicines-10-02720-f005:**
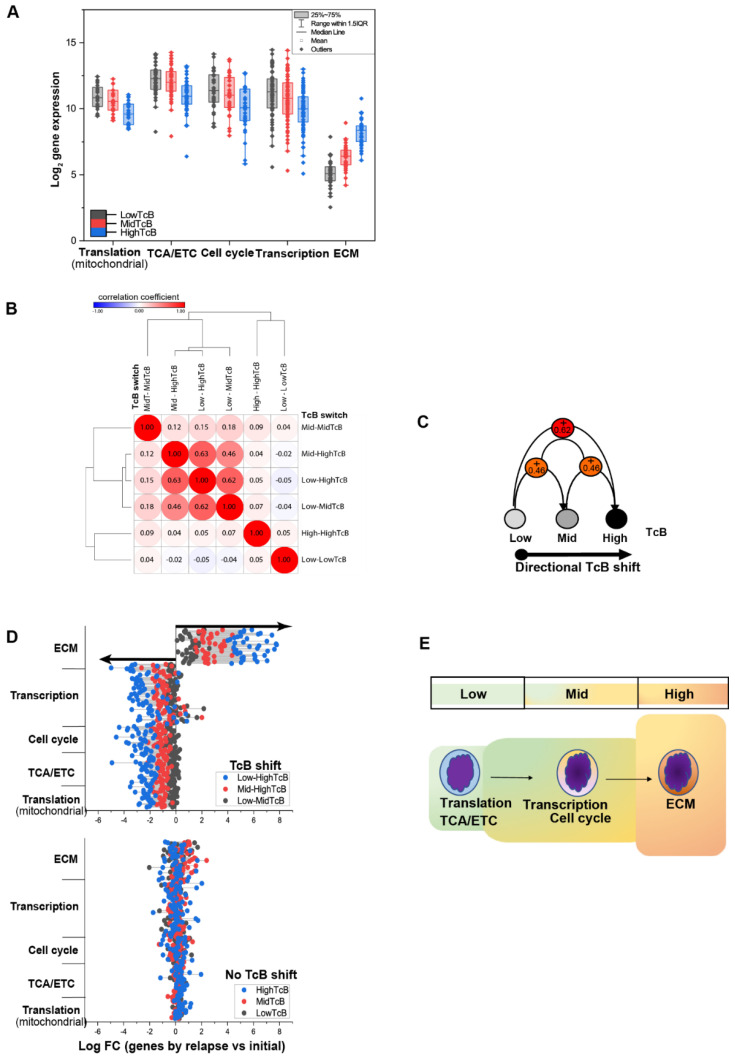
**Biological trajectory of cancer progression in pediatric ALL.** (**A**) The box plot represents log_2_ expression values (y-axis) of genes representing higher-order biological processes (x-axis) enriched by TcB groups. (**B**,**C**) Correlation graph by Spearman rank showing r values of TcB values comparing leukemic relapse with and without TcB shifts indicates a positive correlation in the order of increasing TcB shifts. (**D**) Lollipop plot comparing log_2_ fold changes in gene expression (in x-axis) associated with higher-order processes (y-axis) between relapse and the initial diagnosis is shown with and without TcB shift. (**E**) Schematics illustrating the predicted biological progression path in ALL.

**Figure 6 biomedicines-10-02720-f006:**
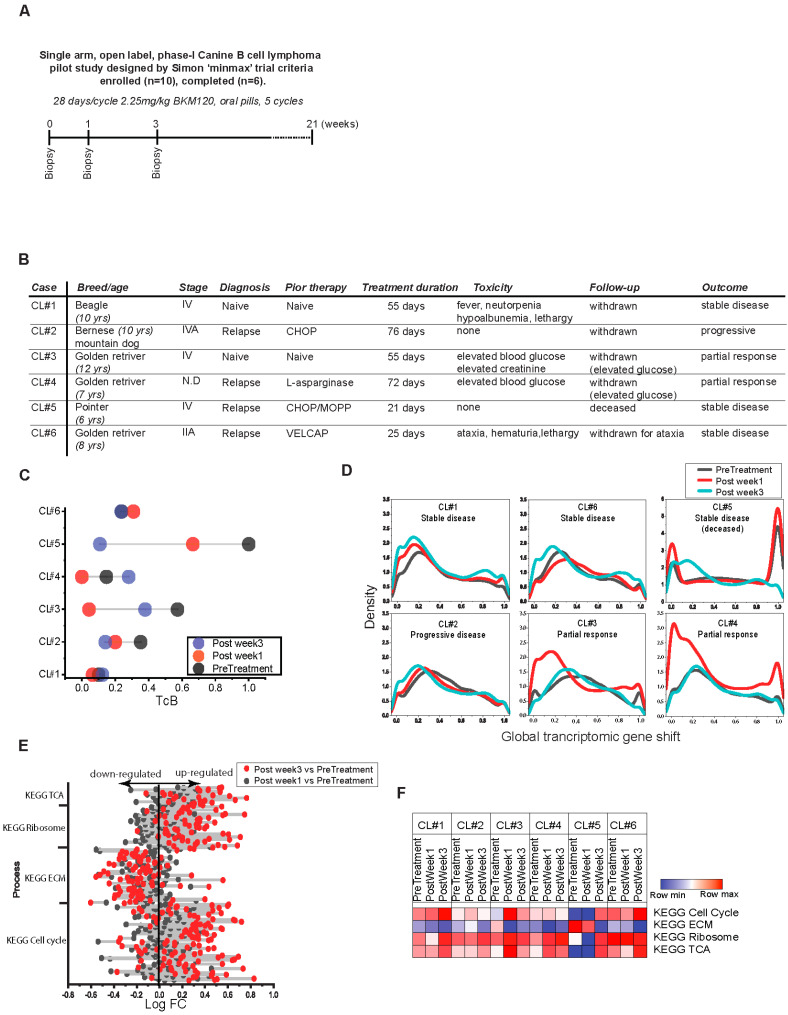
**Targeted therapy impacts the biological trajectory of canine lymphoma.** (**A**) Schematic illustration of BKM120 canine lymphoma clinical trial design. (**B**) Clinical characteristics of canine lymphoma trial subjects including clinical stage, diagnosis, prior therapies (CHOP-cyclophosphamide, hydroxydaunorubcin, oncovin/vincristine, and prednisone; MOPP-mechlorethamine hydrochloride, oncovin, adriamycin/doxorubicin, vincristine, L-asparginase), and treatment durations. (**C** Lollipop plot displaying TcB change by pre-treatment and BKM120 treatment intervals on x-axis with canine lymphoma subjects on y-axis. (**D**) Kernel density plot of global genes (n = 30,311) transformed by median shift for overall gene behavior between pre-treatment and treatment intervals in x-axis indicated by each canine subject treated with BKM120 (**E**) Lollipop plot representing fold change in log_2_ expression of individual genes representing higher-order biological processes (y-axis) comparing BKM120 treatment intervals vs. pre-treatment and in x-axis averaged by canine lymphoma subjects is represented in x-axis. (**F**) Heatmap represents the average log_2_ expression of genes representing higher-order biological processes by individual canines at pre-treatment and BKM120 treatment intervals.

**Figure 7 biomedicines-10-02720-f007:**
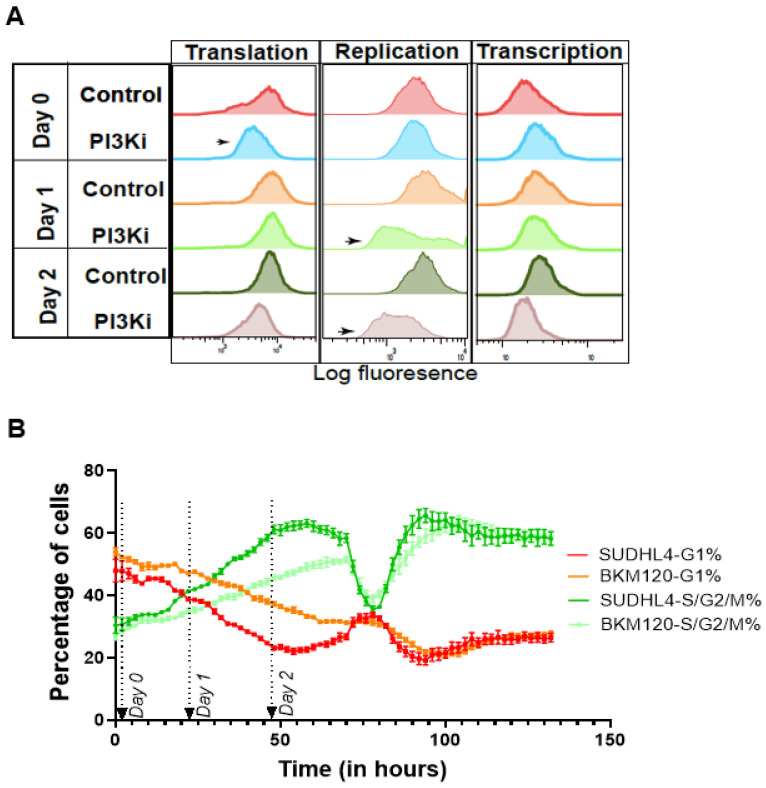
**SUDHL4 cells treated with BKM120 recapitulate the predicted biological sequence of responses.** (**A**) Histograms represent global transcriptional, replication, and translational activity measured in SUDHL4 cells pre-treated and released from BKM120 (1 μM) and quantified as intensity (x-axis) on indicated days using fluorescently labeled Click-IT substrates. (**B**) Line graph shows changes in cell cycle pattern in SUDHL-4 cells transfected with fluorescently labelled cell cycle indicator, represented as percentage of cells in G1, S-G2-M cells and time, following BKM120 release.

## Data Availability

“All data generated from this study are available in the main text or the Appendix A”. Transcriptomic datasets were download from www.C-Bioportal.org, accessed on 1 March 2021 and pediatric solid tumor datasets were downloaded from https://omics-oncogenomics.ccr.cancer.gov/cgi-bin/JK, accessed on 12 November 2021.

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
