# Peer review of "An Analysis of Transcriptomic Burden Identifies Biological Progression Roadmaps for Hematological Malignancies and Solid Tumors"

_biomedicines, 2022, doi:10.3390/biomedicines10112720_

Round 1

Reviewer 1 Report

The subject is somewhat clear, and it has been explored much more than the current introduction gives credit. The article presents a good idea.  Although the initial question is interesting, I have a few issues with the study.

This article fits into the framework of translational medicine: how to fill the gap between basic sciences and clinical sciences. The author can add a paragraph in the introduction to explain the idea in the context of translational medicine. Example of an article should be cited:

doi: 10.1080/03007995.2017.1385450

doi: 10.1007/s00262-014-1645-5
1. Title: it is very general. Reformulate and precise which cancer
2. Abstract: it captures the appropriate essence of the manuscript. Excellent.
3. Introduction: The introduction identifies the problem that is being addressed in the manuscript and develops and states the purpose of the manuscript.
4. Tables and figures: Quality of figures is so important too. Please provide some high-resolution figures. Some figures have a poor resolution.
5. References: I have verified all references and all key references are correct. Please update the reference according to the article proposed above.
6. Methods:

7. Discussion:

* The authors don't discuss the  limitations of the study correctly. Please add it.

* The authors don't compare their study with other studies. Is your roadmaps the best?
8. Conclusion: The conclusion is not justified by the methods and results. Reformulate it
9. There are still some mistakes in grammar and misprints, the authors should carefully check this manuscript.

10. It would be very useful if the authors would make their source code and data

available as supplementary material. This would promote the usage of the proposed

approach and allow others to take better advantage of this research.

* I have enjoyed reading, and I am in favor of publication after suitable.

Author Response

Thanks to the reviewer for taking the time to review our manuscript and appreciating this research and providing us with very thoughtful and constructive comments. Additionally, we are very grateful for giving us the opportunity to revise and resubmit this manuscript. I am pleased to inform you that we have now addressed all concerns in the revised manuscript. The revisions are listed below and are also available in the tracked version of the manuscript. We would like to apologize for the low-resolution images rendered poorly during PDF conversion. We have now embedded higher-resolution images and have also provided the editorial team with the original high-resolution images as well. We hope these revisions will be acceptable for acceptance for publication and will be approved for publication.

Concerns

1. The author can add a paragraph in the introduction to explain the idea in the context of translational medicine. Example of an article should be cited:

doi: 10.1080/03007995.2017.1385450

doi: 10.1007/s00262-014-1645-5

Response: These references are now included in the revised version

The context of translational relevance to medicine are now addressed in 2 parts within the introduction, also included below.

2nd paragraph, lines 53-58

A challenge in advancing precision oncology is the lack of knowledge about the chronological order of biological progression paths taken by tumors.[3]  An imperative aspect of translational science is the generation of knowledge for precision medicine applications through the pooling and analysis of databases on the omics level. This information is available from academic and industrial entities funded by public and private funds.

4th paragraph, lines 100-107

Finally, we provide evidence that this directional biological trajectory is present in malignancy through a prospective drug interventional study involving lymphoma in companion dogs. In this study, we are applying our prior comparative oncology investigations to leverage the translational relevance of lymphoma in dogs and humans. This is increasingly regarded as a valuable strategy for bridging the gap between basic and clinical research.

  1. Title: it is very general. Reformulate and precise which cancer (Revised as suggested)
    2. Abstract: it captures the appropriate essence of the manuscript. Excellent. (Thank you)
    3. Introduction: The introduction identifies the problem that is being addressed in the manuscript and develops and states the purpose of the manuscript.
    4. Tables and figures: Quality of figures is so important too. Please provide some high-resolution figures. Some figures have a poor resolution. (Provided, also high-resolution figures are now uploaded separately), now included
    5. References: I have verified all references and all key references are correct. Please update the reference according to the article proposed above.
    6. Methods:
  2. Discussion:
  • The authors don't discuss the limitations of the study correctly. Please add it. We have now included more descriptions (in discussion starting lines 607-641), and included below.

Our TcB transformation and biological learning strategy has limitations, including a stringent cutoff that eliminated portions of patient for gene-centric analysis. The method must be further developed to accommodate samples representing continuous gradients in transcriptomics. Additionally, samples that don't represent the entire spectrum of tumor progression could skew the results. As an example, our analysis of transcriptomic datasets for head and neck cancers, lung adenocarcinomas, and colon carcinomas detected TcB shifts in either low or high levels, suggesting these samples could represent either early or advanced biological conditions, meaning biological trajectory extraction is not possible. Our analysis pipeline included curation of data that eliminated ambiguous biological processes, such as "disease" or "development," since our intent was to illustrate and collapse enrichments based on biochemical activities directly related to cell proliferation. Several bioinformatic tools are required in order to render biological trajectory data, which is limited to the annotations and interactions defined within the databases. Future directions for improving this methodological strategy include procuring appropriate datasets through large prospective trials, integration of clinical outcome and multi-omic datasets, along with a diagnostic or drug-controlled decision making based on the information of TcB analyses to further solidify its use in actionable cancer care.

* The authors don't compare their study with other studies. Is your roadmaps the best? We have now included more descriptions (in discussion starting from lines 552-583), and included below.

Several studies have attempted to unravel the biological progression roadmaps in cancer. Based on the evolution of genetic changes, Fearon and Vogelstein initially proposed a linear tumor progression sequence. [49] Many studies followed this approach by aligning gene expression datasets with mutational events in order to estimate temporal biological patterns in tumor progression. [48,50,51] Nevertheless, genomic datasets from cross-sectional tumor collection can include samples from unknown disease states, treatment status, environmental exposures, etc., which exhibit tremendous mutation heterogeneity and varying loads of mutational burdens. Therefore, aligning mutations with temporal biological order is considered a weak strategy. [52] Despite mutation assessments being considered sufficient guides for determining and administering effective cures, Vogelstein asserts that a crucial need in basic cancer research is a better understanding of the biological pathway trajectories. [52] There have been further models using probabilistic or Bayesian networks. In these models, genes in one pathway become parents of all genes in the next, and parental genes tied by mutation were integrated into the probability model. [53,54] Lastly, progression at the pathway level was inferred from a priori gene assignment, but only when the pathway had many gene sets. However, none of these approaches were successful in identifying homogenous biological trajectories among cancers, which led to the conclusion that cancer progression is non-linear. Major drawbacks of these approaches include the inability to analyze transcriptomic patterns unbiasedly and the lack of biologically appropriate hypotheses for identifying biological trajectories. Our strategy assumes that transcriptional complexity will continue to increase with tumor progression and the transcriptomic burden will increase. DNA levels in a cell remain relatively constant and can be synthesized in a short period of time. However, RNA levels are higher, so cells need longer periods of time to synthesize enough RNA to divide. Malignant cells may proliferate more rapidly when this constraint is removed, and their RNA content must gradually increase. Our results illustrate this generalized concept with several lines of evidence, including TcB shift from low to high-TCB in ALL, reversibility in canine lymphoma, and transcriptional lag in SUDHL4 cells with BKM120 treatment, all indicative of cancer progression.  

  1. Conclusion: The conclusion is not justified by the methods and results. Reformulate it

We have now included additional data (from SUDHL4 cells) simulating the dynamics of biological trajectory comparing the results from BKM120 treatment in canine lymphoma as Figure 7 and supported the conclusions within discussion Lines 579-583).

  1. There are still some mistakes in grammar and misprints, the authors should carefully check this manuscript. (Corrected)
  2. It would be very useful if the authors would make their source code and data available as supplementary material. This would promote the usage of the proposed approach and allow others to take better advantage of this research. (Please see the supplemental data link, and as stated in Methods (CODE Availability) in Appendix A, R source code and algorithm for adapting this method of analysis have been included already)
  • I have enjoyed reading, and I am in favor of publication after suitable. (Thank you  very much for the appreciation)
  •  
  • Please note, revised version could not be attached with this message, and has been shared with editorial team along with high resolution images.   Ms. Cyrus Cheng E-Mail: [email protected] Assistant Editor

Thank you,

Sincerely,

Ravi Dashnamoorthy

Reviewer 2 Report

Dashnamoorthy Ravi and colleagues present a quality and well-written manuscript describing biological progression roadmaps in cancer identified by transcriptomic burden analysis.

Authors used gene-behavior analysis to evaluate TcB groups to discover biological courses of tumor progression. They found that solid tumors and hematological malignancies (n=4179) share conserved biological patterns, and biological network complexity decreases at increasing TcB levels. An analysis of gene expression datasets including pediatric leukemia patients revealed TcB patterns with biological directionality and survival implications. A prospective interventional study with PI3K targeted therapy in canine lymphomas proved that directional biological responses are dynamic. To conclude, TcB-enriched biological mechanisms detected the existence of biological trajectories within tumors. Using this prognostically informative novel informatics method, which can be applied to tumor transcriptomes and progressive diseases inspires the design of progression-specific therapeutic approaches.

Authors rationalized that the transcriptome of an individual patient at a given interval represents proportion of time spent by tumor in that phase in relation to entire stages of tumor progression. They developed a method providing an independent positional assignment for each patient through weighing their individual transcriptome and integrated gene behavior function across all disease phases.These metrics constitute an abacus-like frame for the gene expression datasets in which patients are ordered by their overall increases in transcriptomic activity and their gene functions into progressive trajectory of biological complexity. The following partitioning of data and bioinformatic analysis defined the biological nature of tumor progression on a global level.

Finally, authors conclude that their analysis of 21 different tumor types composed of 4179 cancer patients provides a blueprint for predicting the causal path of disease progression and potential therapeutic targets.

 Overall, the manuscript is valuable for the scientific community and should be accepted for publication after edits are made.

===========================

Other comments:

1) Please check for typos throughout the manuscript.

2) Line 480. With regards to p53 – authors are kindly encouraged to cite the following article that describes novel therapeutics that target mutant p53.
DOI: 10.1021/acsptsci.2c00164

Author Response

Thanks to the reviewer for taking the time to review our manuscript and appreciating this research and providing us with very thoughtful and constructive comments. Additionally, we are very grateful for giving us the opportunity to revise and resubmit this manuscript. We have fixed the grammatical errors and typos, as suggested. The revised version and high-resolution images have been provided to the editorial team. (We are unable to upload the manuscript with this response)

With regard to Comment 2) Line 480. With regards to p53 – authors are kindly encouraged to cite the following article that describes novel therapeutics that target mutant p53.
DOI: 10.1021/acsptsci.2c00164

As this reference has not been indexed, neither volume nor page numbers are listed. When available, we will do our best to include this interesting reference prior to the final decision. 

Thank you very much for your review. We hope these revisions will be acceptable for acceptance for publication and will be approved for publication.